# A systematic review and meta-analysis for association of *Helicobacter pylori* colonization and celiac disease

**Fazel Isapanah Amlashi**[1], **Zahra Norouzi**[1], **Ahmad Sohrabi**[2,3], **Hesamaddin Shirzad-Aski**[2]*, **Alireza Norouzi**[1], **Ali Ashkbari**[1], **Naghme Gilani**[1], **Seyed Alireza Fatemi**[1], **Sima Besharat**[1,2]*

**1** Golestan Research Center of Gastroenterology and Hepatology, Golestan University of Medical Sciences, Gorgan, Iran, **2** Infectious Diseases Research Center, Golestan University of Medical Sciences, Gorgan, Iran, **3** Cancer Control Research Center, Cancer Control Foundation, Iran University of Medical Sciences, Tehran, Iran

* Shirzad_hessam@yahoo.com (HS-A); s_besharat_gp@yahoo.com (SB)

**Data Availability Statement:** The data underlying the results presented in the study are available from supporting information files.

## Abstract

### Background and objectives

Based on some previous observational studies, there is a theory that suggests a potential relationship between *Helicobacter pylori* (*H. pylori*) colonization and celiac disease (CeD); however, the type of this relationship is still controversial. Therefore, we aimed to conduct a systematic review and meta-analysis to explore all related primary studies to find any possible association between CeD and human *H. pylori* colonization.

### Data sources

Studies were systematically searched and collected from four databases and different types of gray literature to cover all available evidence. After screening, the quality and risk of bias assessment of the selected articles were evaluated.

### Synthesis methods

Meta-analysis calculated pooled odds ratio (OR) on the extracted data. Furthermore, heterogeneity, sensitivity, subgroups, and publication bias analyses were assessed.

### Results

Twenty-six studies were included in this systematic review, with a total of 6001 cases and 135512 control people. The results of meta-analysis on 26 studies showed a significant and negative association between *H. pylori* colonization and CeD (pooled OR = 0.56; 95% CI = 0.45–0.70; P < 0.001), with no publication bias (P = 0.825). The L'Abbé plots also showed a trend of having more *H. pylori* colonization in the control group. Among subgroups, ORs were notably different only when the data were stratified by continents or risk of bias; however, subgroup analysis could not determine the source of heterogeneity.

**Funding:** Golestan University of Medical Sciences, Gorgan, Iran has supported this study (grant number: GOUMS111462). The funders had no role in study design, data collection and analysis, decision to publish, or preparation of the manuscript.

## Conclusions

According to the meta-analysis, this negative association might imply a mild protective role of *H. pylori* against celiac disease. Although this negative association is not strong, it is statistically significant and should be further considered. Further investigations in both molecular and clinic fields with proper methodology and more detailed information are needed to discover more evidence and underlying mechanisms to clear the interactive aspects of *H. pylori* colonization in CeD patients.

## Systematic review registration number (PROSPERO)

CRD42020167730 https://www.crd.york.ac.uk/prospero/display_record.php?RecordID=167730.

## Introduction

Celiac disease (CeD) is an inflammatory autoimmune disease with small intestine enteropathy disorder triggered by gluten ingestion [1]. CeD is a multifactorial disease that has risen from the interaction between gluten and immune, genetic, and environmental factors and can be diagnosed at any age [2]. Serological prevalence of CeD is around 1% in the general population; with a slight difference between Western and Eastern countries (1% vs. 1.6%) [3–5]. The incidence and prevalence of CeD have increased over the years that could be due to better diagnostic tools, screening of high-risk individuals, and general awareness or due to the real increasing rate of the disease based on the alteration of lifestyle or gut microbiota [1,6].

*H. pylori* is one of the most common chronic bacterial infections worldwide, it can cause significant gastroduodenal diseases [7]. The infection affects up to 90% of the population in developing countries and around 40% in developed countries. Generally, half of the global population can have *H. pylori* colonization [8]. There are controversies about the probable relation between *H. pylori* colonization and autoimmune diseases [9]. *H. pylori* can modulate some immune responses and play a protective role against some other diseases such as inflammatory bowel disease (IBD) [10]. Otherwise, in genetically predisposed persons, *H. pylori* can influence immune responses and cause microscopic duodenal inflammation and can be related to more severe damage associated with some gastrointestinal diseases [11–14]. In this regard, some studies reported no relationship between *H. pylori* and CeD [15–17]; whereas, others stated that *H. pylori* can protect against CD [14,18]. There are different hypotheses and justification about the mechanism of interaction between *H. pylori* and CeD.

The hygiene hypothesis is one of the explanations for the protective role of *H. pylori* infection against CeD and claims that lower exposure to some infectious agents in childhood leads to an increase in the risk of autoimmune diseases such as asthma, IBD, allergic rhinitis, and eczema [13,19–22]. Different mechanisms are suggested for this hypothesis. One of them is T helper (Th) 1/Th2 deviation, which suggests that the presence of infection causes the balance of Th1/Th2 to move towards an immunosuppression state. T regulatory (Treg) lymphocytes might be induced or augmented (a phenomenon called bystander suppression) due to infection [19,22,23]. Also, there are more suggested mechanisms for this hypothesis, such as antigenic competition/homeostasis, non-antigenic ligands, and gene-environment interaction [19,22,23].

Another hypothesis is the protective role of *H. pylori* on the modification of gluten digested by proteases secretion, and increase of the pH of the gastric lumen, which reduces the immunogenicity of gluten [13,14,19,22–24]. Some studies suggested another theory and supported the idea that a specific virulence factor, such as CagA in *H. pylori*, may give the bacterium a protective role in CeD. This function of CagA may be due to the association with Treg cells in the immunoregulation mechanism [17,24–26].

The fourth important theory is related to empirical antibiotic treatment in patients suspicious to gastrointestinal bacterial overgrowth, which was eventually diagnosed as CeD. This treatment may lead to the unintentional removal of *H. pylori* [27–29].

Otherwise, some researchers claimed that *H. pylori* may be involved in the development of CeD. Mooney et al. mentioned that maybe *H. pylori* acts as an environmental trigger for CeD, as observed in *Campylobacter* infection, in a study of the US military [21,30]. However, the mechanism of this act is not properly defined. Furthermore, a study at Colombia University showed increased intraepithelial lymphocytes in the duodenal mucosa in patients with *H. pylori* infection that could be reversed by the eradication of *H. pylori* [31].

Up to now and based on our search, no systematic review and meta-analysis have been conducted about this subject. Due to the inconsistency of the previous primary studies, we aimed to prepare a systematic review to scrutinize any possible association between *H. pylori* colonization and CeD.

## Materials and methods

### Eligibility criteria

The design of the present study was based on the methodology of the Preferred Reporting Items for Systematic Reviews and Meta-Analyses (PRISMA) guideline [32]. This review is registered in the international prospective register of systematic reviews (PROSPERO) with registration number CRD42020167730 (Available: https://www.crd.york.ac.uk/prospero/display_record.php?RecordID=167730).

### Definition

**Celiac disease.** CeD patients were defined if they were positive for CeD-specific antibodies (comprise autoantibodies against TG2, including endomysial antibodies (EMA), and antibodies against deamidated forms of gliadin peptides (DGP)), associated with biopsy-proven histopathology or positive genetic tests: HLA-DQ2 or HLA-DQ8 haplotypes in the primary studies.

**Controls.** Based on the definitions in the primary studies, controls consisted of those apparently healthy people or celiac disease was ruled out through negative serology/biopsy.

### Inclusion/Exclusion criteria

Inclusion criteria included: 1) Case-control, cross-sectional, suitable cohort, and brief-report studies that contain evidence about the relationship between *H. pylori* and CeD, 2) Human clinical study, 3) Having at least one parameter of the relative risk effect sizes such as risk ratio (RR), odds ratio (OR), and hazard ratio (HR) or can be calculated based on the information of *H. pylori* status in both case and control groups.

The status of *H. pylori* should be detected by urea breath test (UBT), rapid urease test (RUT), culture, enzyme-linked immunosorbent assay (ELISA), histology, immunohistochemistry (IHC), and Polymerase chain reaction (PCR) methods. Furthermore, the CeD should be

confirmed by endoscopy and biopsy ± serology tests and/or HLA DQ2/DQ8 genotyping. Articles without any of this information were excluded from further analysis.

## Sources and search strategy

All relevant articles, published from 1990 to the end of the second month of 2021 were collected from the electronic database of PubMed, ProQuest, Scopus, and Web of Science. The ProQuest database was also reviewed for related dissertations, manually. Other related protocols were manually searched in PROSPERO. Besides, to expand the scope of the search and find any additional studies, we reviewed the different types of grey literatures such as meeting and conference abstracts for relevant articles (including Digestive Disease Week® (DDW), United European Gastroenterology (UEG), and American College of Gastroenterology). Moreover, we searched manually some key journals in the CeD and *H. pylori* fields, including "World Journal of Gastroenterology", "Journal of Pediatric Gastroenterology and Nutrition", "Gut", "Gastroenterology", and "Helicobacter". In the end, we performed a manual search in the references of the selected articles. We did not impose geographical or linguistic restrictions on our search, and non-English studies with English abstracts were also included.

The Medical subject heading (MeSH) database was used to find various terms of CeD and *H. pylori*. Two main keywords were "*Helicobacter pylori*" and "Celiac". For better searching in the databases, we made syntaxes from a combination of free-text method, MeSH terms, the keywords, and Boolean operators (AND/OR/NOT). Moreover, calculating NNR (number need to read) helped us in evaluating the output of syntax and dedicating. The following syntax was applied in PubMed and adjusted for each search engine based on its search guidelines (Celiac OR "Celiac disease" OR Coeliac OR "Coeliac disease" OR (Disease AND Celiac) OR (Disease AND Coeliac) OR "Gluten Enteropathy" OR (Enteropathy AND Gluten) OR "Gluten Enteropathies" OR (Enteropathies AND Gluten) OR "Gluten-Sensitive Enteropathy" OR (Enteropathy AND Gluten-Sensitive) OR "Gluten-Sensitive Enteropathies" OR (Enteropathies AND Gluten-Sensitive) OR (Sprue AND Celiac) OR (Sprue AND Coeliac) OR (Sprue AND Nontropical) OR "Nontropical Sprue" OR "Celiac Sprue" OR "Coeliac Sprue" OR "Sprue" "Endemic Sprue" OR "Gluten Intolerance") AND ("*Helicobacter pylori*" OR "*Campylobacter pylori*" OR "*Campylobacter pyloridis*" OR "*Campylobacter pyloris*" OR "*Helicobacter nemestrinae*" OR "*Helicobacter* infection" OR "HP infection" OR "HP infections" OR "*Helicobacter* infections" OR "*Helicobacter* colonization" OR "HP colonization" OR "*Helicobacter* colonization" OR "*H Pylori*" OR "enterohepatic *Helicobacter*" OR "EHS" OR "*Helicobacter*") AND 1990/01/01:2021/02/20[dp].

The search outputs were exported into the Endnote software (Version X7; Thompson Reuters Corporation, Toronto, ON, Canada) to remove any duplicate and screen articles. The first step of screening for determining the eligible primary articles was done based on the titles and abstracts. Two independent reviewers (A.A. and A.F.) evaluated the selected full-text articles and separately classified them into three relevant, irrelevant, and unsure groups, based on the eligibility criteria. Any discrepancy was supervised by a third reviewer (S.B.) and resolved via a consensus. If there was still a doubt in the selection of a study, the whole team made the final decision.

## Quality and risk of bias assessment

The quality and risk of bias assessment of the selected articles were independently evaluated by two reviewers (Z.N. and N.G.), using the standard checklist of the Newcastle-Ottawa Scale (NOS) form [33]. The studies were classified into three poor, fair, and good categories based on getting a score between zero and eight in the selection, comparability, and outcome/exposure domains based on the standard guideline. Consensus and the opinion of the third reviewer (S.B) resolved any disagreements between the two reviewers.

## Data extraction

Two authors (Z.N, F.A) independently performed data extraction from each paper based on a defined protocol. The data were divided into three categories and included 1) General information: the first author's names, year of publication, journal names, country, and region; 2) The risk of bias assessment; 3) Study setting: study design, study duration, sample size and total population recruited in each study, inclusion and exclusion criteria, age group and age range, sex, the definition of CeD, source of the data, diagnostic methods, OR and 95% CI, and the ethical approval. In cases of any missing or additional necessary data, we further sent an e-mail to each author, in charge of the related study.

## Data synthesis and analysis

We used the R software environment, version 4.0.2, and the "meta" package to calculate pooled OR and 95% CIs for each study. Based on the methodological heterogeneity between the studies, the random-effect model (REM) was used for the combination method. The size of the combined effect was calculated and displayed in a forest plot. The standard chi-square test ($Q$ Cochrane test) and $I^2$ scale evaluated the heterogeneity between studies. Combinations with an $I^2$ scale equal or more than 70% assumed as a severe heterogeneity. In addition, a Drapery plot was drawn to show the P-value functions for the included studies, as well as pooled estimates, and to provide two-sided confidence intervals for all possible alpha levels (confidence interval function) [34,35]. For finding the cause(s) of heterogeneity, subgroup and sensitivity analyses were used. The studies were stratified based on each factor, including the median of publication year, overall age group of the participants, continent and regions of the studies, *H. pylori* detection method, the risk of bias assessment, as well as sampling quality. In the last subgroup, sampling, the articles were categorized into two groups (I) "Appropriate sampling" or (II) "Not appropriate sampling" groups. The studies were located in the "Not appropriate sampling" group, if the number of participants in each case and control group was less or equal than 50 people or the ratio of control to the cases was more than four times or less than 90%.

The funnel chart, followed by Begg's test analyzed possible publication bias. Baujat and L'Abbé plots were used to show the effect of each study on the heterogeneity and overall influence of the results [36–38]. Finally, a one-out remove method was used to detect the influence of each study on the overall pooled OR.

## Results

### Search results

In total, the general searching step obtained 1732 papers. After removing duplicates, 1106 studies were screened, based on their titles and abstracts. Among them, 1014 articles were excluded due to not meeting the inclusion criteria. Therefore, the full-texts of 92 remained articles were completely evaluated. Finally, 26 studies were included in the systematic review and also meta-analysis (kappa (κ) = 0.84) [11,13–18,21,24–27,39–52], and the others were excluded due to a reason mentioned in Fig 1 (Fig 1). In the end, there was no suitable cohort study in the included studies.

### Characteristics of the included articles

The designs of all the included studies were case-control, with a total of 6001 cases and 135512 control people. Among the studies Lebwohl *et al*. had the highest study population with 130308 participants (included 2689 cases and 127619 controls) [14]. Most studies

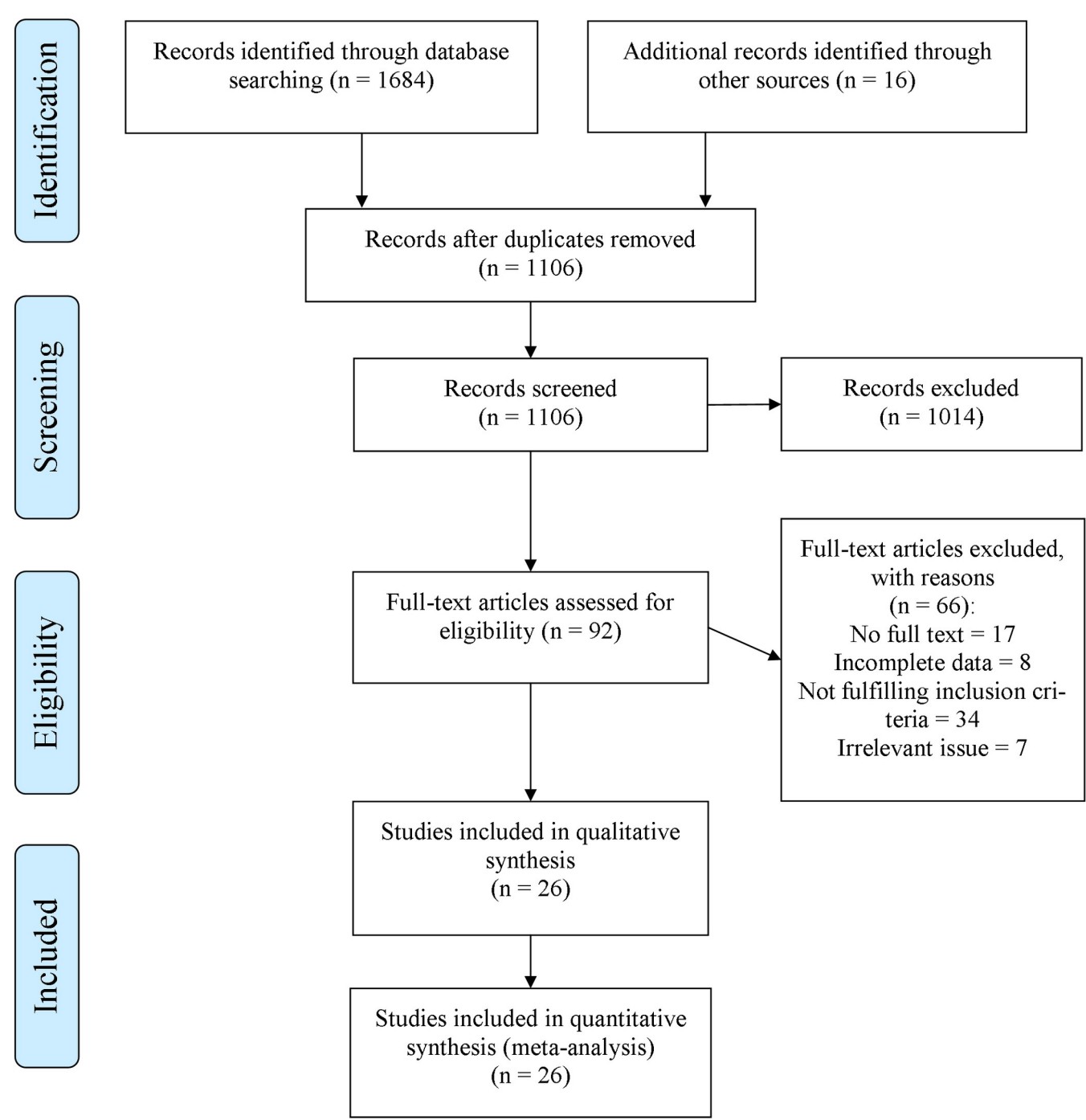

**Fig 1. Flow diagram of literature search, screening, and selection of studies for review and meta-analysis.**

were from Turkey (N = 6) [15–18,51,52] and Italy (N = 5) [11,27,44,45,48]. Some studies had been performed only on children [13,15–17,21,44,47,51,52] or adults [18,25,27,39, 43,45,48], the others evaluated both age groups [11,14,42,49,50]. Table 1 shows the details of the included articles. The NOS assessment showed that only 19.2% of the studies had good quality.

**Table 1. Main characteristics of the included studies in the meta-analysis on the *Helicobacter pylori* infection and celiac disease (CeD).**

| First author | Year | Journal title | Country | Definition of CeD | Risk of bias assessment | Case-number | Control—number | Laboratory tests of *H. pylori* |
|---|---|---|---|---|---|---|---|---|
| Karttunen | 1990 | Journal of Clinical Pathology | Finland | Registry | F[c] | 27 | 27 | IHC[f] |
| Crabtree | 1992 | Journal of Clinical Pathology | England | Endoscopy + Serology + Positive response to GFD[b] | P[d] | 99 | 250 | ELISA[g] |
| Feeley | 1998 | Journal of Clinical Pathology | Ireland | Registry | F | 70 | 70 | IHC |
| Diamantti | 1998 | Intestinal Disorders (poster) | Argentina | Endoscopy | P | 108 | 113 | IHC |
| Diamantti | 1999 | The American Journal of Gastroenterology | Argentina | Endoscopy + Serology + Positive response to GFD | P | 104 | 75 | IHC |
| Luzza | 1999 | Journal of Pediatric Gastroenterology and Nutrition | Italy | Endoscopy+ Positive response to GFD | G[e] | 81 | 81 | IHC + RUT[h] + ELISA |
| Konturek | 2000 | The American Journal of Gastroenterology | Germany | Endoscopy + Serology | F | 91 | 40 | ELISA |
| Ciacci | 2000 | European Journal of Gastroenterology and Hepatology | Italy | Endoscopy | P | 187 | 76 | IHC + ELISA |
| Dettori | 2001 | Digestive and Liver Disease (Poster) | Italy | Endoscopy + Serology | G | 152 | 306 | UBT[i] + IHC |
| Aydogdu | 2008 | Scandinavian Journal of Gastroenterology | Turkey | Endoscopy + Serology | P | 96 | 235 | IHC |
| Rostami-Nejad | 2009 | Revista espanola de enfermedades digestivas | Iran | Endoscopy + Serology | P | 28 | 422 | IHC |
| Rostami-Nejad | 2011 | Archives of Iranian Medicine | Iran | Endoscopy + Serology | P | 24 | 226 | IHC |
| Lebwohl | 2013 | American Journal of Epidemiology | American | Endoscopy | F | 2689 | 127619 | IHC |
| Simondi | 2015 | Clinics and Research in Hepatology and Gastroenterology | Italy | Endoscopy + Serology + HLA DQ2/DQ8 genotyping | F | 73 | 404 | UBT + IHC |
| Jozefczuk | 2015 | European Review for Medical and Pharmacological Science | Poland | Endoscopy + Serology | F | 74 | 296 | UBT |
| Lasa | 2015 | Arquivos de Gastroenterologia | Argentina | Endoscopy + Serology | G | 72 | 240 | IHC |
| Jozefczuk | 2016 | European Review for Medical and Pharmacological Science | Poland | Endoscopy + Serology | F | 76 | 84 | UBT + ELISA |
| Uyanikoglu | 2016 | Euroasian Journal of Hepato-Gastroenterology | Turkey | Endoscopy | F | 31 | 592 | IHC |
| Narang | 2017 | Journal of Gastroenterology and Hepatology | India | Endoscopy + Serology + Positive response to GFD | G | 324 | 322 | IHC + RUT |
| Broide | 2017 | AGA abstracts | Germany | Endoscopy + Serology | F | 50 | 50 | UBT + IHC |
| Lucero | 2017 | Frontiers in Cellular and Infection Microbiology | Chile | Endoscopy + Serology + HLA DQ2/DQ8 genotyping | G | 66 | 50 | IHC + RUT |
| Agin | 2019 | Archives of Medical Science | Turkey | Endoscopy + Serology | P | 256 | 1012 | IHC |
| Cam | 2018 | EHMSG [a] | Turkey | Endoscopy | P | 141 | 150 | UBT + IHC |
| Dore | 2018 | Helicobacter | Italy | Endoscopy + HLA DQ2/DQ8 genotyping | F | 270 | 127 | UBT + IHC |
| Bayrak | 2020 | Helicobacter | Turkey | Endoscopy + Serology | F | 482 | 2060 | UBT + IHC |
| Bayrak | 2021 | Digestive Diseases | Turkey | Endoscopy + Serology | F | 278 | 505 | UBT + IHC |

[a]XXXIst International Workshop on *Helicobacter* & Microbiota in Inflammation & Cancer;

[b]gluten-free diet;

[c]Fair;

[d]Poor;

[e]Good;

[f]Immunohistochemistry;

[g]Enzyme-linked immunosorbent assay;

[h]Rapid Urease Test;

[i]Urease Breath Test.

## Association between *H. pylori* and CeD

By analyzing a combination of 26 studies that addressed the association between *H. pylori* colonization and CeD, a negative association was found (pooled OR = 0.56; 95% CI = 0.45 − 0.70; P ≤ 0.001). The REM and Chi-squared method showed a significant statistical heterogeneity between the studies $I^2$ = 81%, P ≤ 0.001, and $X^2$ = 130, P ≤ 0.001, respectively (Fig 2).

Fig 3A shows a Drapery plot, presenting that the estimated effect is less than 1 in the majority of studies, similarly as shown in the Funnel plot. As the horizontal dashed lines show the CIs for common alpha levels (0.1, 0.05, 0.01, 0.001), at any level of prediction interval (90, 95, 99, 99.9%), the main triangle line (the red-bolded line in the online version) did not intersect the null effect line (Incidence Rate Ratio (IRR) = 1). According to the results of the Funnel plot and Begg's tests, there was no significant publication bias (P = 0.825, Fig 3B). The Baujat plot (Fig 3C) showed that the study of Narang *et al.* contributes more than the other studies on the overall heterogeneity and influences the pooled effect size. The L'Abbé plots (Fig 3D) also showed a trend of having more *H. pylori* colonization in the control group.

## Subgroup and sensitivity analyses

The heterogeneity between studies was evaluated by multiple subgroup analyses in the six subgroups. Based on the continents of the studies, there were few studies in Asia, which made the results unreliable for the statistical analysis [13,49,50]. Europe has a higher OR (OR = 0.63, 95% CI = 0.49 − 0.82) than America (OR = 0.55, 95% CI = 0.40 − 0.76) and Asia (OR = 0.23,

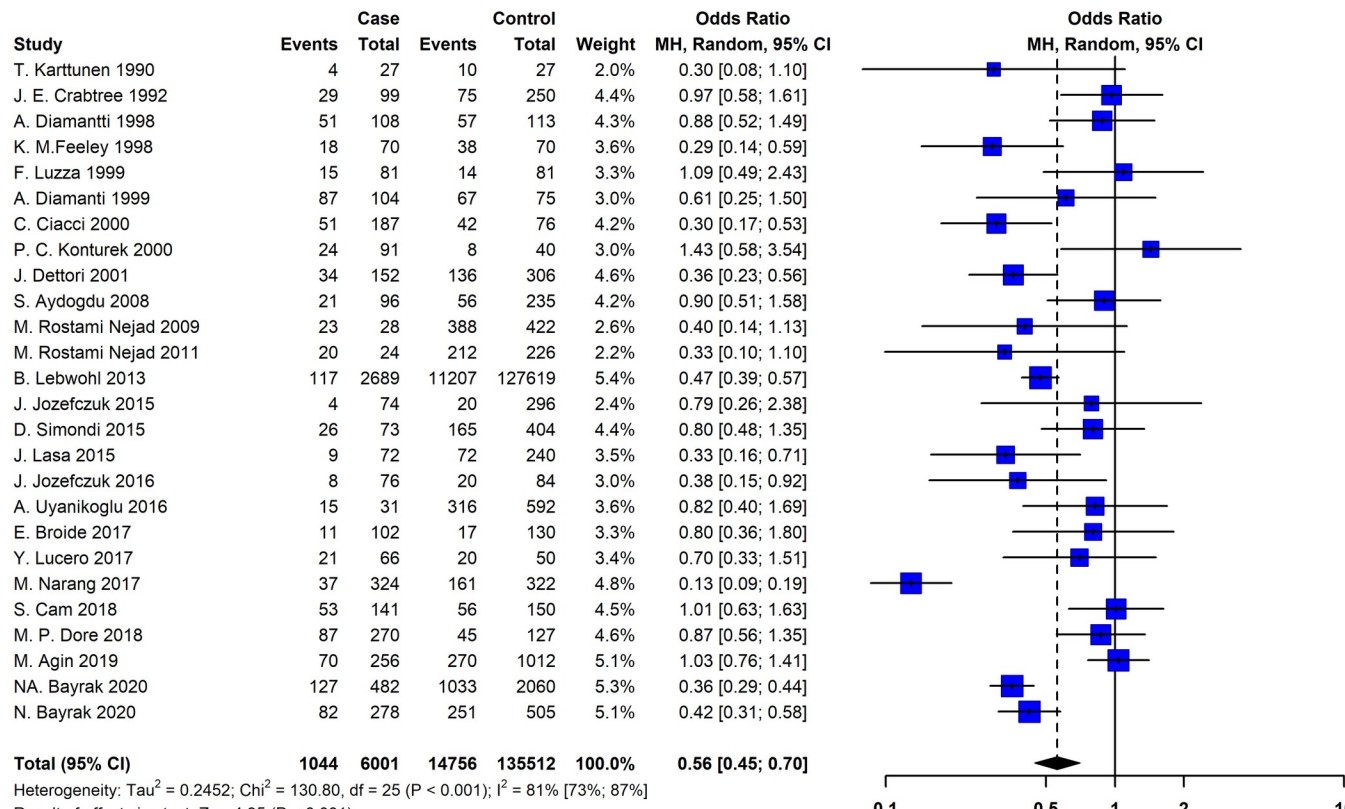

**Fig 2. The Forest plot of the meta-analysis on the association of the *Helicobacter pylori* and celiac disease.** The random-effects model was used to calculate pooled ORs with 95% CI. CI, Confidence interval; OR, Odds ratio.

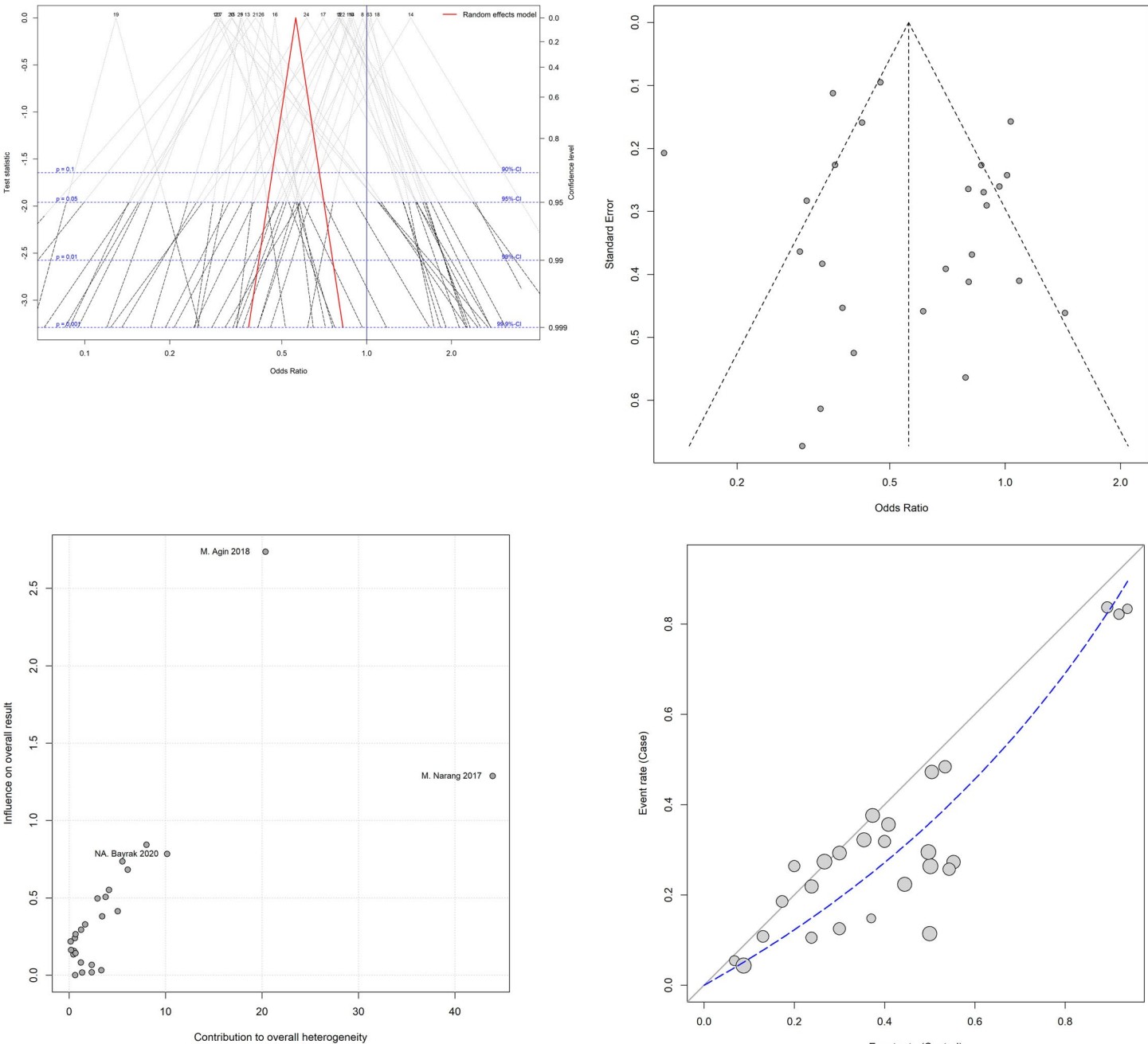

**Fig 3. (A)** The drapery plot shows p-value functions for the included studies as well as pooled estimates and the two-sided confidence intervals for all possible alpha levels (confidence interval function). Each number represents a study. The red-bolded line (in the online version; bold line in the printed version) indicates the range of pooled odds ratio (OR) in each alpha level. **(B)** The Funnel chart shows no significant publication bias, as all studies are in a symmetric scheme. **(C)** The Baujat plot shows the effect of each study on the heterogeneity and overall influence of the results. As can be seen in the graph, the study of Narang et al. contributes more than the other studies to the overall heterogeneity. Each circle represents a study. **(D)** The L'Abbé plot shows more positive results for H. pylori colonization in the control group. The diagonal (x = y) oblique line represents the odds ratio (OR) equal to one. The dashed line indicates the OR of the studies. The size of each circle indicates the assigned random weight of each study. As the picture shows, most studies show an OR of less than one.

95% CI = 0.10 − 0.52). However, further evaluation is needed due to the low rate of studies, especially in Asia. Other subgroups could not determine the source of heterogeneity (Table 2).

**Table 2. Different subgroup analysis by stratifying the data.**

| Subgroups | | N[a] of articles | N of cases | N of controls | OR[b] | P-Value | Chi-Square | I[2] | Tau[2] |
|---|---|---|---|---|---|---|---|---|---|
| Combination of all studies | | 26 | 6001 | 135512 | 0.56 | <0.001 | 130.80 | 81 | 0.2452 |
| Year | Before 2014 | 13 | 3756 | 129540 | 0.56 | 0.001 | 32.47 | 63 | 0.1350 |
| | 2014 & after 2014 | 13 | 2245 | 5972 | 0.56 | <0.001 | 98.17 | 88 | 0.3804 |
| Overall age | Children (Less than 18) | 10 | 1910 | 4875 | 0.57 | <0.001 | 92.64 | 90 | 0.4505 |
| | Adult (More than 18) | 9 | 894 | 1919 | 0.60 | 0.004 | 22.59 | 65 | 0.1785 |
| | All age | 7 | 3197 | 128718 | 0.52 | 0.08 | 11.41 | 47 | 0.0734 |
| Continent | Europe | 18 | 2586 | 6445 | 0.63 | <0.001 | 75.43 | 77 | 0.2138 |
| | America | 5 | 3039 | 128097 | 0.55 | 0.15 | 6.81 | 41 | 0.0516 |
| | Asia | 3 | 376 | 970 | 0.23 | 0.06 | 5.67 | 65 | 0.3481 |
| Sampling | Appropriate sampling | 14 | 1929 | 3794 | 0.57 | <0.001 | 97.94 | 87 | 0.4459 |
| | Inappropriate sampling | 12 | 4072 | 131718 | 0.55 | 0.001 | 30.2 | 64 | 0.0983 |
| *H. Pylori* detection tests | IHC[c] & ELISA[d] | 20 | 5307 | 134275 | 0.58 | <0.001 | 77.88 | 76 | 0.1598 |
| | UBT[e] & RUT[f] | 6 | 694 | 1234 | 0.52 | <0.001 | 45.88 | 89 | 0.9442 |
| Risk of bias | Poor | 9 | 1043 | 2559 | 0.72 | 0.009 | 20.23 | 60.0 | 0.1231 |
| | Fair | 12 | 4263 | 131954 | 0.55 | 0.001 | 31 | 65.0 | 0.0873 |
| | Good | 5 | 695 | 999 | 0.39 | <0.001 | 31.84 | 87.0 | 0.6152 |

[a]Number;

[b]Odds ratio;

[c]Immunohistochemistry;

[d]Enzyme-linked immunosorbent assay;

[e]Urease Breath Test;

[f]Rapid Urease Test.

Based on the one-out removed method, the removal of any of the studies could not significantly affect the pooled results. The sensitivity analysis was used to find possible outlier studies with more influence on the results. In this regard, the studies with poor design or not appropriate sampling (the most influential factors in the methodology) were omitted, and the overall effect size and level of heterogeneity were recalculated. The overall OR decreased from 0.56 (95% CI = $0.45 − 0.76$) to 0.40 (95% CI: $0.26 − 0.62$) but no change in the heterogeneity was observed. The results of the sensitivity analysis confirmed that the correct methodology was significantly associated with a lower amount of overall OR (Table 3).

## Discussion

Most information about the relationship between *H. pylori* colonization and CeD is inferred from case-control studies and more prospective cohort and /or systematic review are needed to provide the strongest evidence about this association. In this regard, the present study is the first systematic review and meta-analysis that aimed to clarify any possible association between *H. pylori* colonization and CeD. The meta-analysis showed a mild negative relationship between *H. pylori* and CeD, and confirmed the findings of some previous studies, showing *H. pylori* was significantly more prevalent in apparently healthy people than celiac patients [11,13,14].

The protective effect of *H. pylori* on some allergic and atopic diseases and other inflammatory diseases could be considered as starting point of the debates about the association between CeD and *H. pylori* [10,17,18,21].

For clearing the possible protective effect of the immune-suppressive mechanisms of *H. pylori* on CeD, the first step is the specification of the chronological order of CeD and *H. pylori*

**Table 3. Subgroups sensitivity analysis.**

| Models | Subgroups | N[a] of articles | N of cases | N of controls | OR[b] | P-Value | Chi-Square | I² | Tau² |
|---|---|---|---|---|---|---|---|---|---|
| All studies | | 26 | 6001 | 135512 | 0.56 | <0.001 | 130.80 | 81 | 0.2452 |
| Without "poor" quality studies | | 17 | 4958 | 132953 | 0.49 | <0.001 | 75.27 | 79 | 0.1973 |
| | Europe | 13 | 1807 | 4722 | **0.57[†]** | <0.001 | 38.58 | 67 | 0.1433 |
| | America | 3 | 2827 | 127909 | **0.47** | 0.40 | 1.84 | 0 | 0 |
| | Asia | 1 | 324 | 322 | **0.13** | NA | NA | NA | NA |
| | Appropriate sampling | 9 | 1229 | 2034 | 0.40 | <0.001 | 38.76 | 79 | 0.3288 |
| | Inappropriate sampling | 8 | 3729 | 130919 | 0.62 | <0.001 | 26.81 | 74 | 0.1195 |
| Without "not appropriate sampling" studies | | 14 | 1929 | 3794 | 0.54 | <0.001 | 97.94 | 87 | 0.4459 |
| | Poor | 5 | 700 | 1760 | **0.98** | 0.98 | 0.4 | 0 | 0 |
| | Fair | 5 | 600 | 1085 | **0.45** | 0.32 | 4.67 | 14 | 0.0200 |
| | Good | 4 | 629 | 949 | **0.34** | <0.001 | 26.14 | 89 | 0.6170 |
| | Europe | 11 | 1425 | 3119 | **0.66** | <0.001 | 37.91 | 74 | 0.1925 |
| | America | 2 | 180 | 353 | **0.56** | 0.04 | 4.33 | 77 | 0.3658 |
| | Asia | 1 | 324 | 322 | **0.13** | NA | NA | NA | NA |
| Without "poor" quality and "not appropriate sampling" studies | | 9 | 1229 | 2034 | 0.40 | <0.001 | 38.76 | 79 | 0.3288 |
| | Fair | 5 | 600 | 1085 | 0.45 | 0.32 | 4.67 | 14 | 0.0200 |
| | Good | 4 | 629 | 949 | **0.34** | <0.001 | 26.14 | 89 | 0.6170 |
| | Europe | 7 | 833 | 1472 | **0.48** | 0.11 | 10.3 | 42 | 0.0691 |
| | America | 1 | 72 | 240 | **0.33** | NA | NA | NA | NA |
| | Asia | 1 | 324 | 322 | **0.13** | NA | NA | NA | NA |

[a]number;

[b]odds ratio;

†The bolded items show a statistical difference in OR of region subgroup.

colonization. As there was not any cohort study on this subject, there is no strong evidence to show a cause and effect relationship between these two and a protective effect of *H. pylori* on the occurrence of CeD. Therefore, conducting a cohort study should be considered as a top priority in further investigations of this subject.

Beyond our subgroup analyses, there are more interactive aspects of the pathophysiology of CeD and pathogenesis of *H. pylori* that could help us to explore the causes of the heterogeneity. Both CeD and *H. pylori* cause inflammation and immune response in the GI tract and also *H. pylori* could escape human immune defense via different mechanisms that suppress both innate and adaptive immune response [53]. *H. pylori* can evade host immune response via activating signaling pathway, sustain intracellularly in macrophages and induction of apoptosis, inhibition of leukocyte migration, and producing arginase enzyme.

Recent studies reported that Vacuolating cytotoxin (VacA) of *H. pylori* has an immunomodulatory role by inhibiting integrin-linked kinase (ILK) [54]. VacA also intervenes with the antigen presentation of MHC class II [55] and it directly suppresses T-cells rather than antigen presenting cells (APC) [56]. It is possible that these suppressive mechanisms produced by *H. pylori*, affect the autoimmune response to gluten and reduce the level of gluten enteropathy in CeD. Considering the notion of a balance between immune suppressive mechanisms of *H. pylori* and autoimmune response in CeD, it seems more probable to observe a protective effect against autoimmune response among patients who are infected with a type of *H. pylori*.

In this systematic review and meta-analysis related to the association of *H. pylori* colonization and CeD, our search strategy prepared a comprehensive source of articles that contains any study with related evidence. Subgroup and sensitivity analyses showed that two factors (sampling method and the quality of article) could be more effective and important for case-control studies and it is better to be addressed in future studies. In addition, the analyses showed that this protective effect of *H. pylori* on CeD was more observed in Asian countries. After removing the studies with poor quality and non-appropriate sampling, the pooled OR was still statistically lower in a study from Asia, in comparison to both studies from Europe and America. However, heterogeneity was still more than 70% in this continent subgroup. Unfortunately, after removing the studies in sensitivity analysis, only one study from Asia remained, and the obtained results could not completely determine the source of heterogeneity. Here, we suggest conducting well-designed studies in Asia, especially in the East Asian population to clarify the role of ethnicity in the association of *H. pylori* colonization and CeD.

The present study had some limitations. First, just a few numbers of the selected articles had a good-quality method and many of them did not consider many confounding factors in the design of the study such as March type, the severity of celiac, *H. pylori* virulence, and IgA deficiency that could affect the results. Second, based on the results, the heterogeneity of outcomes was high, and the subgroup analysis could not recognize the source of it completely. Third, most of the studies were conducted in Western countries, especially in Turkey and Italy. There was no article from Africa. Moreover, only three articles reported the situation in Asia. Studying other ethnicities can be a suitable target for further investigations.

## Conclusion

The meta-analysis of 26 clinical studies included 6001 CeD patients from both children and adult age-groups showed a mild protective role of *H. pylori* against celiac disease. The main obstacle on the way of placing the role of *H. pylori* in CeD under scrutiny is the shortage of accurate details in the clinical studies. To clear the relationship between *H. pylori* and CeD, it is suggested to implement cohort studies and also further primary studies with good methodology. Extending the scope of study about this subject to the molecular context may prepare more answers to the current controversies.

## Supporting information

**S1 Checklist.**
(DOC)

**S1 File.**
(DOCX)

## Author Contributions

**Conceptualization:** Fazel Isapanah Amlashi, Zahra Norouzi, Hesamaddin Shirzad-Aski, Alireza Norouzi, Ali Ashkbari, Naghme Gilani, Sima Besharat.

**Data curation:** Fazel Isapanah Amlashi, Zahra Norouzi, Hesamaddin Shirzad-Aski, Alireza Norouzi, Ali Ashkbari, Naghme Gilani, Seyed Alireza Fatemi, Sima Besharat.

**Formal analysis:** Ahmad Sohrabi.

**Investigation:** Zahra Norouzi, Ahmad Sohrabi, Hesamaddin Shirzad-Aski, Alireza Norouzi, Ali Ashkbari, Naghme Gilani, Seyed Alireza Fatemi, Sima Besharat.

**Methodology:** Ahmad Sohrabi, Hesamaddin Shirzad-Aski, Alireza Norouzi, Sima Besharat.

**Project administration:** Hesamaddin Shirzad-Aski, Sima Besharat.

**Software:** Ahmad Sohrabi.

**Supervision:** Hesamaddin Shirzad-Aski, Alireza Norouzi, Sima Besharat.

**Validation:** Zahra Norouzi, Hesamaddin Shirzad-Aski, Seyed Alireza Fatemi, Sima Besharat.

**Visualization:** Ahmad Sohrabi.

**Writing – original draft:** Fazel Isapanah Amlashi, Zahra Norouzi, Ahmad Sohrabi, Hesamaddin Shirzad-Aski, Ali Ashkbari, Naghme Gilani, Seyed Alireza Fatemi, Sima Besharat.

**Writing – review & editing:** Fazel Isapanah Amlashi, Zahra Norouzi, Hesamaddin Shirzad-Aski, Alireza Norouzi, Sima Besharat.

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
