## [Decision Letter · Decision Letter 0]

7 Jan 2021

PONE-D-20-31127

A Systematic Review and Meta-analysis for Association of Helicobacter pylori Colonization and Celiac Disease

PLOS ONE

Dear Dr. Besharat,

Thank you for submitting your manuscript to PLOS ONE. After careful consideration, we feel that it has merit but does not fully meet PLOS ONE’s publication criteria as it currently stands. Therefore, we invite you to submit a revised version of the manuscript that addresses the points raised during the review process.

Your manuscript has been reviewed by two experts in the field and their comments are appended below. We believe the raised comments would improve the manuscript and may allow a revised version to be published in *PLOS ONE*.

Regarding my own concern, it is important that you revise the Introduction and Discussion sections in order to make the text more precise and concise. In the Methods section, the inclusion criteria comprise articles published until the “end of 2019”; authors should update their search to include more recent studies (if available), and the exact date of the updated search must be specified in the text. The first paragraph presented in “Quality and risk of bias assessment” should be reallocated to “Sources and search strategy”. It is unclear from the Methods section why the authors used a modified checklist of the Newcastle-Ottawa Scale (NOS) since there is a validated scale of NOS for case-control studies without the need for changes. What was modified in the used version? My last concern is related to the eligibility criteria: why cohort studies were not included?

The reviewers have constructively detailed other relevant issues. Please try to address all issues carefully in a revised manuscript.

We look forward to receiving your revised manuscript.

Kind regards,

Endi Lanza Galvão

Academic Editor

PLOS ONE

Journal Requirements:

Reviewers' comments:

Reviewer's Responses to Questions

**Comments to the Author**

1. Is the manuscript technically sound, and do the data support the conclusions?

Reviewer #1: Yes

Reviewer #2: Yes

2. Has the statistical analysis been performed appropriately and rigorously? 

Reviewer #1: Yes

Reviewer #2: Yes

3. Have the authors made all data underlying the findings in their manuscript fully available?

Reviewer #1: Yes

Reviewer #2: Yes

4. Is the manuscript presented in an intelligible fashion and written in standard English?

Reviewer #1: Yes

Reviewer #2: Yes

5. Review Comments to the Author

Reviewer #1: Review of article:

A Systematic Review and Meta-analysis for Association of Helicobacter pylori

Colonization and Celiac Disease

1. Well-written manuscript fora n area with significant H. pylori infection prevalence.

2. A meta-analysis of a large number of studies and a total of 5241 patients and >13000 controls.

3. A clear method.

4. Introduction section:

- Please use a more recent and widely accepted reference. Ref no 2, 6 or 10 may be changed to for example: the ESsCD guideline 2019. Al-Toma et al. United European Gastroenterol J . 2019 Jun;7(5):583-613.

- The introduction can be shortened: for example the paragraph on the modes of presentation may be deleted or shortened. Also that on complications and diagnosis.

We need to focus on epidemiology, pathogenesis and relation with HP.

5. Discussion: This a very extensive section and difficult to comprehend. More than 2000 words.

This section should be more concise. The authors are trying to tackle all aspect of the topic. That is good; but the section is very difficult to read and in my opinion the data on earlier publications can be extremely shortened. All hypotheses , theories and mechanisms can be much summarized to give a clear message to the reader.

6. Conclusion in abstract and main body of manuscript: The suggestion that H. pylori has protective role towards CD needs to be somewhat modified. We do not know if the weak negative association may be regarded as protective. I would suggest saying ( The negative association might imply a mild protective role of HP against celiac disease.]

6. General note: CD is used widely as an abbreviation for celiac disease. Recently there is a trend to use CeD as an abbreviation to differentiate celiac from Crohn disease. Therefore, I would advise the authors to change to CeD.

7. The references are Ok. Some editing is needed due to software problems (can be done manually).

Reviewer #2: In the present systematic review and metaanalysis paper, the authors have described an association between colonisation of H pylori and Celiac disease. The authors have found a negative association between H pylori and Celiac disease. There a marked heterogeneity in the studies. Of 24 studies included, only 7 of them shows an association (inconsistency)

Comments:

1. Introduction: There reason and hypothesis of an association should be reflected in the introduction. The introduction is too lengthy and mostly unfocussed (the paragraph 2 may be deleted and paragraph 3 may be shorted).

2. Definition used in the study: The definition (reference) of celiac disease is not clear. Further how was a diagnosis of H pylori infection was made in the studies. Was the infection considered positive if one or two tests were positive.

3. Who were the controls?

4. Of 7 studies showing an association between celiac disease and IHC, the basis of diagnosis H pylori in 6 of them was IHC. Is this also a factor that can explain the heterogeneity.

5. Discussion is very exhaustive and unfocussed. This should shortened substantially.

Minor:

Legends may be written separately, at the end of the manuscript.

6. PLOS authors have the option to publish the peer review history of their article (what does this mean?). If published, this will include your full peer review and any attached files.

Reviewer #1: No

Reviewer #2: **Yes: **Dr Govind Makharia

---

## [Author Response · Author response to Decision Letter 0]

4 Feb 2021

3-Feb-2021

Dear Prof. Joerg Heber,

The editor in chief of the journal of PLOS One,

We appreciate the time and efforts of the editor and reviewers in reviewing this manuscript. We revised the manuscript (ID: No. PONE-D-20-31127, entitled "A Systematic Review and Meta-analysis for Association of Helicobacter pylori Colonization and Celiac Disease") and corrected all points mentioned by the journal’s reviewers. All changes based on the editor and reviewer’s comments and questions are marked with the highlighted tracked changes in the “track-changed version” of the revised manuscript text. A “clean version” of the revised manuscript has also been prepared. Besides, in the following lines, we answered the editor and reviewer’s questions, marked by a bullet.

The editor’s comment:

Regarding my own concern, it is important that you revise the Introduction and Discussion sections in order to make the text more precise and concise.

• Thank you for the suggestion. Based on your concern, as well as reviewers’ comments, both Introduction and Discussion sections were revised. We try to concise the sections. In addition to summarizing, we tried to write the text in a coherent and integrated manner. Some parts in the introduction related to the modes of presentation and diagnosis of celiac disease were deleted. Furthermore, some parts of the discussion were transferred to the introduction.

In the Methods section, the inclusion criteria comprise articles published until the “end of 2019”; authors should update their search to include more recent studies (if available), and the exact date of the updated search must be specified in the text.

• Thank you for your suggestion that improves our result. We updated our systematic search until 28 February 2021. With the new syntax, 59 articles were added and after the screen step, five of them were selected for full-text evaluation. Finally, two articles (with 760 cases and 2565 control population) were added to the systematic review and meta-analysis part. Both new papers belonged to Turkey. According to the new results, all figures were also updated.

The first paragraph presented in “Quality and risk of bias assessment” should be reallocated to “Sources and search strategy”.

• We greatly appreciate your suggestion. The mentioned first paragraph reallocated to the “Sources and search strategy” part.

It is unclear from the Methods section why the authors used a modified checklist of the Newcastle-Ottawa Scale (NOS) since there is a validated scale of NOS for case-control studies without the need for changes. What was modified in the used version?

• Thank you for your very detailed point. We intended to make changes to the standard form of NOS, if necessary, for evaluation of a particular type of cohort studies, but in the end, there was no need to change. So we used the standard form. But (unfortunately), we forgot to delete the “modified” word from our first protocol plan and the text of the manuscript. As a result and based on your insightful comment, we deleted the word “modified” (which was written wrongly) before the name of the Newcastle-Ottawa Scale (NOS) form. 

My last concern is related to the eligibility criteria: why cohort studies were not included?

• We apologize for the ambiguity. At first, we wanted to evaluate any case-control and all type of cohort study. But after evaluation of the full text of studies, no cohort study passed our inclusion criteria based on our aims. So that, we removed the name of “the cohort study type” from the method section. However, to avoid ambiguity and also based on your comment, we have again added more explanation about not finding cohort studies in the methods and result parts. We added the following sentence to the method part: “Case-control, cross-sectional, suitable cohort, and brief-report studies that contain evidence about the relationship between H. pylori and CD”. Furthermore, we added the following sentence to the result part: “In the end, there was no suitable cohort study in the included studies”. Thank you for your question.

The answers to the comments of reviewer 1:

1. Well-written manuscript for an area with significant H. pylori infection prevalence. 2. A meta-analysis of a large number of studies and a total of 5241 patients and >13000 controls. 3. A clear method.

• Thank you. We greatly appreciate the reviewer’s efforts to carefully review the paper.

4. Introduction section: Please use a more recent and widely accepted reference. Ref no 2, 6 or 10 may be changed to for example: the ESsCD guideline 2019. Al-Toma et al. United European Gastroenterol J . 2019 Jun;7(5):583-613.

• We greatly appreciate the reviewer’s efforts to carefully review the paper. According to the opinion of the reviewer, we used the mentioned reference (ESsCD guideline) and added it to the introduction of the manuscript.

The introduction can be shortened: for example, the paragraph on the modes of presentation may be deleted or shortened, also that on complications and diagnosis. We need to focus on epidemiology, pathogenesis and relation with HP.

• We greatly appreciate the reviewer’s efforts to carefully review the paper. According to the opinion of the reviewer, we did the following changes in the introduction of the manuscript. We try to concise the sections. In addition to summarizing, we tried to write the text in a coherent and integrated manner. Some parts in the introduction related to the modes of presentation and diagnosis of celiac disease (CeD) were deleted. We also added information about the prevalence of CeD in the world (epidemiology information) and the relation of CeD with H. pylori and its possible mechanisms to this part.

5. Discussion: This is a very extensive section and difficult to comprehend, more than 2000 words. This section should be more concise. The authors are trying to tackle all aspect of the topic. That is good; but the section is very difficult to read and in my opinion the data on earlier publications can be extremely shortened. All hypotheses, theories and mechanisms can be much summarized to give a clear message to the reader.

• We appreciate the reviewer’s comment. We tried to concise the sections. In addition to summarizing, we tried to write the text in a coherent and integrated manner. All hypotheses, theories, and mechanisms were summarized at first and transferred to the introduction part, as were more related to this part (based on the opinion of another reviewer). All changes in this part are marked with the highlighted tracked changes.

6. Conclusion in abstract and main body of manuscript: The suggestion that H. pylori has protective role towards CD needs to be somewhat modified. We do not know if the weak negative association may be regarded as protective. I would suggest saying (The negative association might imply a mild protective role of H. pylori against celiac disease).

• Thank you for the suggestion. We considered your comment and used the “mild” term for the negative association and the protective possibility.

6. General note: CD is used widely as an abbreviation for celiac disease. Recently there is a trend to use CeD as an abbreviation to differentiate celiac from Crohn’s disease. Therefore, I would advise the authors to change to CeD.

• According to the insightful suggestion of the reviewer, we updated the abbreviation of “celiac disease” from “CD” to “CeD”, to differentiate celiac from Crohn’s disease and make the text smoother for the readers.

7. The references are Ok. Some editing is needed due to software problems (can be done manually).

• After performing the plain text, the references part was checked, revised, and corrected.

The answers to the comments of reviewer 2:

In the present systematic review and metaanalysis paper, the authors have described an association between colonisation of H pylori and Celiac disease. The authors have found a negative association between H. pylori and Celiac disease. There are a marked heterogeneity in the studies. Of 24 studies included, only 7 of them shows an association (inconsistency).

• Thank you, we tried to evaluate any possible source for the heterogeneity. We also revised our conclusion with the following sentence: The negative association might imply a mild protective role of H. pylori against celiac disease.

Comments:

1. Introduction: There reason and hypothesis of an association should be reflected in the introduction. The introduction is too lengthy and mostly unfocussed (the paragraph 2 may be deleted and paragraph 3 may be shorted).

• We greatly appreciate the reviewer’s efforts to carefully review the paper. According to the opinion of the referee, we tried to concise the sections and write the text in a coherent and integrated manner. In addition to summarizing, we deleted the mentioned paragraphs. We transferred some part of the discussion (reasons and hypotheses) to the introduction part. All changes in this part are marked with the highlighted tracked changes.

2. Definition used in the study: The definition (reference) of celiac disease is not clear. Further how was a diagnosis of H. pylori infection was made in the studies. Was the infection considered positive if one or two tests were positive?

• Thank you for the suggestion. We added the definition section to the method parts and tried to explain the case and control groups. CeD patients were defined if they were positive for CeD-specific antibodies (comprise autoantibodies against TG2, including endomysial antibodies (EMA), and antibodies against deamidated forms of gliadin peptides (DGP)), associated with biopsy-proven histopathology or positive for genetic tests including HLA-DQ2 or HLA-DQ8 haplotypes in the primary studies. 

• We considered H. pylori infection in the study (case/control) population if one of the following tests was positive: urea breath test (UBT), rapid urease test (RUT), culture, enzyme-linked immunosorbent assay (ELISA), histology, immunohistochemistry (IHC), and Polymerase chain reaction (PCR) methods. However, after data extraction, as some tests were used only one or two times in the included studies, we combined similar tests for subgroup analysis to perform statistical analysis. 

3. Who were the controls?

• Based on the definitions in the primary studies, controls consisted of those apparently healthy people, or celiac disease was ruled out through negative serology/biopsy in the control population. We added this definition to the method part.

4. Of 7 studies showing an association between celiac disease and IHC (H. pylori), the basis of diagnosis H. pylori in 6 of them was IHC. Is this also a factor that can explain the heterogeneity?

• Thank you for the insightful comment. We evaluated the included studies based on the detection method of H. pylori by subgrouping to analyze the source of heterogeneity. Among 26 studies, 19 studies used IHC for the detection of H. pylori, among them seven studies (36.8%) had an association between CeD and H. pylori. Otherwise, the other seven studies used some other methods such as RUT and among them, two studies (28.5%) showed the mentioned association. Therefore, it seems that this factor could not play role in heterogeneity (the difference is not statistical).

5. Discussion is very exhaustive and unfocussed. This should be shortened substantially.

• Thank you for the comment. Regarding the comments of both reviewers, we summarized the discussion section. We transferred all hypotheses, and mechanisms that supported both sides of the association to the introduction. We tried to focus on the main result of the present study and discuss more related to the source of heterogeneity and methodology of the included studies.

Minor:

Legends may be written separately, at the end of the manuscript.

• Thank you for the suggestion. According to the guideline of the journal, all legends related to Figures and Tables should be placed in the proper location of the text of the manuscript. Therefore, we stick to the journal guideline.

Best.

---

## [Decision Letter · Decision Letter 1]

15 Feb 2021

A Systematic Review and Meta-analysis for Association of Helicobacter pylori Colonization and Celiac Disease

PONE-D-20-31127R1

Dear Dr. Besharat,

We’re pleased to inform you that your manuscript has been judged scientifically suitable for publication and will be formally accepted for publication once it meets all outstanding technical requirements.

Kind regards,

Endi Lanza Galvão

Academic Editor

PLOS ONE

Additional Editor Comments (optional):

Reviewers' comments:

Reviewer's Responses to Questions

**Comments to the Author**

1. If the authors have adequately addressed your comments raised in a previous round of review and you feel that this manuscript is now acceptable for publication, you may indicate that here to bypass the “Comments to the Author” section, enter your conflict of interest statement in the “Confidential to Editor” section, and submit your "Accept" recommendation.

Reviewer #1: All comments have been addressed

Reviewer #2: All comments have been addressed

2. Is the manuscript technically sound, and do the data support the conclusions?

Reviewer #1: Yes

Reviewer #2: Yes

3. Has the statistical analysis been performed appropriately and rigorously? 

Reviewer #1: Yes

Reviewer #2: Yes

4. Have the authors made all data underlying the findings in their manuscript fully available?

Reviewer #1: Yes

Reviewer #2: (No Response)

5. Is the manuscript presented in an intelligible fashion and written in standard English?

Reviewer #1: Yes

Reviewer #2: Yes

6. Review Comments to the Author

Reviewer #1: (No Response)

Reviewer #2: n the present systematic review and metaanalysis paper, the authors have described an association between colonisation of H pylori and Celiac disease.

Thank you so very much for responding to our comments

7. PLOS authors have the option to publish the peer review history of their article (what does this mean?). If published, this will include your full peer review and any attached files.

Reviewer #1: No

Reviewer #2: **Yes: **Govind K Makharia

---

## [Editor Report · Acceptance letter]

18 Feb 2021

PONE-D-20-31127R1 

A Systematic Review and Meta-analysis for Association of *Helicobacter pylori* Colonization and Celiac Disease 

Dear Dr. Besharat:

I'm pleased to inform you that your manuscript has been deemed suitable for publication in PLOS ONE. Congratulations! Your manuscript is now with our production department. 

Kind regards, 

on behalf of

Dr. Endi Lanza Galvão 

Academic Editor

PLOS ONE